# Into the Dark: Exploring the Deep Ocean with Single-Virus Genomics

**DOI:** 10.3390/v14071589

**Published:** 2022-07-21

**Authors:** Francisco Martinez-Hernandez, Oscar Fornas, Manuel Martinez-Garcia

**Affiliations:** 1Department of Physiology, Genetics and Microbiology, University of Alicante, 03690 Alicante, Spain; franmh@ua.es; 2Centre for Genomic Regulation (CRG), The Barcelona Institute for Science and Technology (BIST), PRBB Building, 08003 Barcelona, Spain; oscar.fornas@upf.edu

**Keywords:** single-virus genomics, deep ocean, marine viruses

## Abstract

Single-virus genomics (SVGs) has been successfully applied to ocean surface samples allowing the discovery of widespread dominant viruses overlooked for years by metagenomics, such as the uncultured virus vSAG 37-F6 infecting the ubiquitous *Pelagibacter* spp. In SVGs, one uncultured virus at a time is sorted from the environmental sample, whole-genome amplified, and sequenced. Here, we have applied SVGs to deep-ocean samples (200–4000 m depth) from global Malaspina and MEDIMAX expeditions, demonstrating the feasibility of this method in deep-ocean samples. A total of 1328 virus-like particles were sorted from the North Atlantic Ocean, the deep Mediterranean Sea, and the Pacific Ocean oxygen minimum zone (OMZ). For this proof of concept, sixty single viruses were selected at random for sequencing. Genome annotation identified 27 of these genomes as bona fide viruses, and detected three auxiliary metabolic genes involved in nucleotide biosynthesis and sugar metabolism. Massive protein profile analysis confirmed that these viruses represented novel viral groups not present in databases. Although they were not previously assembled by viromics, global fragment recruitment analysis showed a conserved profile of relative abundance of these viruses in all analyzed samples spanning different oceans. Altogether, these results reveal the feasibility in using SVGs in this vast environment to unveil the genomes of relevant viruses.

## 1. Introduction

Viruses have a deep impact on microbial populations in all ecosystems. They directly regulate the microbial abundance via cell host lysis [1,2], have a direct effect on primary and secondary production through the viral shunt [3,4], generate microbial diversity through horizontal gene transfer [5], and even hijack the microbial metabolism by means of auxiliary metabolic genes (AMGs) [6,7,8]. Over the last years, the advances in viral ecology have broadened the knowledge on viral diversity, structure, and biogeography [9]. Nevertheless, the full extent of diversity and microdiversity of the viral populations is yet to be discovered, with a potential impact on biogeochemical cycles, highlighting the importance of continuous exploring the virosphere [10].

The marine environment is one of the best-studied habitats in virology due to its direct impact on planetary biogeochemical cycles. Albeit, most of these surveys, because of their feasibility, have been carried out in epipelagic zones [11]. So far, exploring the vast deep ocean below 1000 m depth representing nearly 65% of the Earth’s surface is yet a challenge [12]. Remarkably, the mesopelagic zone (200 to 1000 m depth) and the deep ocean harbor more than half of the marine viral stock [13,14]. Recent studies on these habitats have provided some insights into viral composition by sequencing and discovering novel viral genomes [15,16]. Nevertheless, these results have revealed the fact that our knowledge about deep-ocean viral diversity is incipient [15,16].

Viromics (i.e., viral metagenomics) has been the most commonly employed method to unveil the deep-ocean virosphere, increasing our current knowledge on the global viral diversity [6,11,17], abundance [18], and virus–host interaction [19,20]. Nevertheless, it has been demonstrated that this valuable approach has some drawbacks. For instance, a large sample volume (order of liters) is required, and in some cases, the recovery of microdiverse viral populations (diversity within the same viral species) could be hindered by inherent problems during metagenomic assembly [10,18,21]. Due to this fact, complementary methodologies have been employed to complement viromics. For instance, the use of fosmid libraries [22] allowed the discovery of 99 bathypelagic viral contigs from two Mediterranean Sea sampling points (1000 and 3000 m depth). Another powerful approach has been SVGs [18,23,24], which sorts and sequences the genome of one virus at a time directly from as low as 1 mL of seawater, or even less [18]. SVGs has been employed to reveal novel highly microdiverse abundant viruses not only from marine samples but also from human salivary samples [25]. In a pioneer marine SVG survey, three bathypelagic viral single amplified genomes (vSAGs) were sequenced from the North Atlantic obtained during the Malaspina expedition (4000 m depth) [18]. Using a similar technology as SVGs, albeit not at the level of single viruses, a study using fluorescence-activated virus sorting (FAVS) sequenced a pool of sorted viruses from the North Atlantic Deep Water (~2500 m) and the Antarctic Bottom Water (~4000 m) [26].

Here, we use SVGs in bathypelagic North Atlantic Ocean, the deep Mediterranean Sea, and the oxygen minimum zone (OMZ) in the Pacific Ocean, recovering 27 abundant deep-ocean viruses, demonstrating the potential of this technology to recover representative viruses from the deep-ocean virosphere from very little starting sample material.

## 2. Materials and Methods

### 2.1. Sample Collection

Samples were collected from 4 marine stations during the Malaspina and MEDIMAX expeditions (Figure 1 and Table 1). From the global Malaspina expedition, samples were recovered from the North Pacific Ocean (Station 114, 14°31′43″ N 118°46′26″ W) on 27 May 2011, at 200 m depth, representing the OMZ. Another two samples were collected at 4000 m depth from the North Atlantic Ocean (stations 131 (17°25′39″ N 59°49′43″ W) and 134 (18°19′38″ N 52°38′20.15″ W)) on 26 and 29 November, respectively. The MEDIMAX expedition sample was obtained from the western Mediterranean Sea (37°21′13″ N 00°17′10″ W) at 2000 m depth on 16 October 2015.

All samples were immediately filtered through 0.22 μm polyethersulfone (PES) membrane filters. For the Malaspina expedition samples, after filtration, 100 μL of GlyTE buffer (20 mL 10× TE pH 8.0, 60 mL deionized water, and 100 mL molecular-grade glycerol) was added to 1 mL of bathypelagic seawater, and then the sample was cryopreserved at −80 °C until use. For the MEDIMAX expedition samples, 150 mL of seawater was concentrated to approximately 15 mL using tangential flow filtration system Vivaflow 200 (Sartorius) with 100 KDa PES filter. Concentrated seawater was filtered again by 0.2 μm PES syringe filter, and ultraconcentrated to a final volume of 1.5 mL using the centrifugal filter Amicon Ultra-15 (Millipore, Burligton, MA, USA). Then, the sample was conserved at 4 °C, due to the short time lapse between sampling and viral sorting.

### 2.2. Fluorescence-Activated Viral Sorting and Whole-Genome Amplification

Detailed procedures for sorting, whole-genome amplification, and preparation have been previously described [18,27]. Briefly, after washing, resuspended samples (500 μL) were stained for 20 min in the dark at room temperature, adding 2.5 μL of 1000× SYBR Gold (Invitrogen, Waltham, MA, USA). Then, the excess of SYBR Gold dye was removed by washing samples three times with 500 μL of 0.02 μm-filtered 1X TE buffer, and finally, concentrated stained samples were resuspended in 500 μL of this buffer. Viral sorting (Figure 1) was carried out in a BD Influx sorter (Becton Dickinson, Franklin Lakes, NJ, USA) as described in [18]. Single viruses and pools of viruses (n = 10 and 20, as controls) were sorted in 384-well plates containing 0.6 μL of 16 h UV-radiated and 0.02 μm-filtered 1X TE buffer, and plates were preserved at −80 °C until use. Previous to the amplification, to release the genetic material, viral capsids were lysed by a combination of temperature and pH shocks. For this, 384-multiwell plates were sequentially immersed, one time, in liquid N_2_ and in a 45 °C water bath, for 30 s each. Then, 0.7 μL of lysis buffer D2 (Qiagen) was added for each well, and after 5 min, pH was neutralized employing 0.7 μL of Stop solution (Qiagen). Then, whole-genome amplification was carried out by multiple displacement amplification (MDA) using the Phi29 DNA polymerase (New England Biolab, Ipswich, MA, USA) for 16 h at 30 °C [18]. Real-time whole-genome amplification was monitored by fluorescence thanks to SYTO-9 dye that was added to the MDA reaction (see details in [18]), allowing the identification of positive vSAG amplification during real-time MDA. Then, MDA products were diluted 50-fold in TE buffer.

### 2.3. Sequencing, Viral Identification, and Fragment Recruitment Analysis

A total of 60 out of 317 successfully amplified vSAGs (~20%) were further selected for Illumina sequencing using the Nextera XT DNA library in a MiSeq sequencer (2 × 250 bp) according to the manufacturer’s protocol. Reads were trimmed using trimmomatic [28] (trimmomatic-x.xx.jar SE -phred33 amplicons_seqX_zoneX.fastq.gz amplicons_seqX_sampleX._trimmed.fastq.gz LEADING:3 TRAILING:3 SLIDINGWINDOW:4:20 MINLEN: 150), and assembled with SPAdes (CITA) using k-mers (-k) 33,55,77,99,127, single-cell MDA data (--sc) data option, and careful (--careful) mode.

To identify confident viral contigs, the viral sequence identification standard operating procedure (SOP) with VirSorter2 V.2 from protocols.io (https://dx.doi.org/10.17504/protocols.io.btv8nn9w (accessed on 10 May 2022)) was employed, combining VirSorter2, CheckV, and DRAMv, following the standard procedure for all the assembled contigs longer than 2500 bp. Annotated proteins from the identified viral contigs were compared with the IMG/VR database v3, containing the proteins from 2,332,702 viral genomes, using BLASTP with a cutoff of amino acid similarity >80% and query coverage >90%. Additionally, to detect the presence of similar viruses within the novel vSAGs, all these proteins were compared in an all-versus-all protein BLASTP, using the same above-described BLASTP cutoffs.

Global abundance of the recovered vSAGs was estimated in silico by a metagenomic and viromic fragment recruitment. This analysis was performed as previously described in [18], but employing a less-restrictive nucleotide identity threshold (nucleotide identity cutoff ≥80%) in order to estimate the abundance of the viral populations at the genus level. Viromes from the South Pacific, North Pacific, and North Atlantic bathypelagic regions during the Malaspina expedition (from 3008–4018 m depth) used here for viromic fragment recruitment have been previously reported [6] (see Appendix A). In addition, two metagenomes from the Mediterranean Sea (Adriatic Sea (1000 m depth) and the Ionic Sea (3000 m depth) [22] were also employed to estimate the relative viral abundance and potential activity in the cell fractions.

## 3. Results

For this study, a total of 1328 virus-like particles were sorted by fluorescence-activated viral sorting (FAVS) from the four samples located in the Mediterranean Sea (2000 m depth), North Atlantic (4000 m depth), and North Pacific (OMZ, 200 m depth) Oceans, during the MEDIMAX and Malaspina expeditions (Figure 1, Table 1). Real-time MDA allowed the efficient whole-genome amplification (WGA) of nearly one-quarter (23.87%) of the total sorted single viruses (Table 1). The maximum efficiency of this amplification was obtained in the Mediterranean Sea samples (48.80% of single sorted particles), while the minimum was obtained for the North Atlantic Ocean samples (5.12% of sorted viral particles; station MSP131, 4000 m depth). To check the feasibility of the SVGs method to sort and amplify the genomes of single viruses from long-preserved deep-ocean samples, we randomly selected a representative vSAG population (n = 60 vSAGs, ~20%) of these viral single amplified genomes (vSAGs) for Illumina sequencing from all analyzed bathypelagic samples (Table 1). After assembling, 138 contigs longer than 2500 bp were obtained from 54 of these sequenced vSAGs (Appendix A). As each vSAG is sequenced and assembled separately, different contigs can be assigned to the corresponding vSAG. The largest contig for each genome was between 3356 and 37,461 bp, with a mean length of 16,081 bp per genome (Appendix A). Bioinformatic tools using VirSorter2, CheckV, and DRAMv programs along with a manual curation allowed us to confidently identify 27 of these assembled genomes as bona fide viruses (Table 1) containing terminases, phage lysozymes, or structural viral proteins among other common viral proteins (Appendix A). Remarkably, three AMGs were also found, such as a glycosyl transferase (vSAG 27-M11), and two genes involved in the biosynthesis of the inosine monophosphate (vSAG 27-M11) and the pyrimidine deoxyribonucleotide (vSAG 30-J17) (Appendix A). Unfortunately, protein annotation and clustering did not allow a proper and confident taxonomic classification for most of the viruses mainly due to their novelty and divergence as discussed below. As rather conservative and strict criteria were used in order to identify bona fide viruses (see methods for details), several sequenced vSAGs did not show known viral hallmark genes; thus, they were not finally characterized as viral genomes.

To check the novelty of the bona fide viruses (n = 27), their protein profile was compared with the comprehensive viral database IMG/VR v3.0 (n = 2,332,702 viral genomes) [9]. This analysis (Figure 2) confirmed these vSAGs as novel viruses, not sequenced before by any other methodology. Most of them shared (amino acid similarity >80% and query coverage >90%) less than 15% of their proteins with previously known viruses (mean = 6%, Figure 2A, Appendix A). vSAG 86-3-K18 (length = 13,776 bp) obtained from 2000 m depth in the Mediterranean Sea was the unique vSAG that exceeded this value (mean = 38% of shared proteins) (Figure 2A). We found that the most similar previously known virus in databases was a viral contig obtained from a Mediterranean Sea virome (3511 m depth, DOE’s Joint Genome Institute scaffold ID: Ga0114898_1000086), sharing 60% of the vSAG 88-3-J13 genome (from the North Atlantic Malaspina station 131, 4000 m depth) (Figure 2B). The closest viruses for the rest of the vSAGs showed a lower percentage of shared proteins (see data in Figure 2B and Appendix A for each closest relative virus and vSAG pair).

To ascertain the relativeness between the recovered vSAGs from the different analyzed samples, a pairwise comparison of all vSAGs was performed. The protein-sharing (amino acid similarity >80% and query coverage >90%) analysis showed 14 links (presence of similar proteins) between two vSAGs, and one triple interaction (Figure 3, Appendix A). The highest similarities were found between the vSAGs 86-3-K18 and 86-3-D22 from the MEDIMAX expedition. Genome alignment showed a perfect alignment and synteny (100% nucleotide identity) between different contigs of both viruses (Appendix A), revealing that both correspond to the same virus. Similar results were found for vSAGs 88-3-H13 and 88-3-J13 from the North Atlantic Ocean (MSP131 sampling point), which also exhibited a perfect gene synteny and nucleotide identity (Figure 3, Appendix A).

To analyze the abundance of the vSAGs in the global deep-ocean virosphere, we performed a metagenomic and viromic fragment recruitment, using metagenomes from the deep Mediterranean Sea (600 and 3200 m depth) [22] and from the Malaspina expedition [6]. A nucleotide identity cutoff ≥80% was employed to target similar viruses, putatively within the same genus [10,18]. A homogeneous profile of relative abundance of analyzed vSAGs was observed in the different deep-ocean viromes (Figure 4). Overall, those vSAGs obtained from the North Atlantic (station MSP131) were the most abundant, and in particular, vSAG 88-3-B13 and 88-3-G20 obtained the highest recruitment rates in all analyzed viromes (Figure 4). This higher abundance of both viruses was also observed in the two Mediterranean Sea metagenomes used in this analysis, although significantly lower abundance values for all the vSAGs were observed in these cell metagenomes.

## 4. Discussion

SVGs allows the recovery of thousands of viral particles from low sample volume (in the order of mL) [18], which makes this approach suitable for precious biological samples in which the available amount of sample is limiting. In addition, SVGs can also be effectively applied to samples with low concentration of viruses. In this work, with only 1 mL from Malaspina expedition samples, we were able to sort hundreds of VLPs from bathypelagic regions, where viral load was one order of magnitude less than in surface seawater [13]. The whole-genome amplification (WGA) success rate was, as a mean, ~24%, slightly higher than those values obtained for surface seawater in a previous SVGs survey (17.22%) [18]. Variation in this rate was observed between the different samples, probably generated by the different preservation methods (Table 1) as previously discussed [29]. The best WGA successful rates were obtained from a fresh sample (Mediterranean Sea; WGA success rate = 48.80%) while lower values were obtained in cryopreserved samples with the cryoprotectant glycerol Tris-EDTA (GlyTE) (WGA success rate = 15.56%), commonly used in single-cell genomics. This variation in the WGA success rate is in accordance with a previous SVG benchmarking study, in which the efficiency of the SVGs methodology was tested using different preservation methods [29]. Nevertheless, results of this work demonstrate the ability of SVGs to efficiently recover the genome from long-preserved samples (sorting and amplification performed ~4 years apart from sampling). Likely, the use of fresh unfixed samples, along with short periods of time from sampling to sorting, guarantee a better genome recovery in SVGs [29].

In this pilot SVG survey of the deep-ocean environment, we have identified 25 novel viruses (two of them were independently retrieved and captured in two different vSAGs) (Table 1, Appendix A). The application of SVGs in surface seawater in a prior study highlighted the limitation of viromics to unveil the genetic diversity of microdiverse viral populations [18,21], which play a major role in the microbial biosphere [10]. In this way, the current state of the art of SVGs should be understood as a complementary tool to other existing viromic methodologies, allowing the recovery of key pieces from the complex virosphere jigsaw. The refinement of the viral sorting and sequencing methodologies and large-scale analysis will show the real scope of SVGs in the deciphering of the global virosphere [24].

In addition to these 27 vSAGs identified as confident viral contigs, other amplified genomes (n = 33) were also sequenced with an uncertain origin (Table 1, Appendix A). The lack of a universal conserved viral gene makes the unequivocal identification of genomes as viruses a challenge. These other amplified genomes were not finally considered after performing a refined analysis based on VirSorter2 [30], CheckV, and vDRAM programs that analyze the potential viral origin, quality, and gene annotation of assembled contigs [31,32]. Despite the advances of these in silico tools, some novel bona fide viral genomes could have been discarded here, due to no similarities with current databases and a lack of known viral hallmark genes. The main reason for this is that these tools are constrained by the current knowledge of the virosphere. Nevertheless, we have preferred to choose a conservative approach. Recent works have showed that the deep-ocean viral communities are highly underrepresented in the databases [15,16], which makes the proper identification and characterization of novel viruses very difficult. Another important factor during the viral identification is the presence of microbial genes in the viral genome. Some viruses contain host genes that directly regulate the microbial metabolism, favoring the infective process, (i.e., AMGs [33]), and, for instance, AMGs have been described in deep-ocean viruses [34] as well. Here, we found two viral genomes containing proteins related to the purine and pyrimidine metabolism (Appendix A). It has been reported that viral genes involved in these metabolism pathways are among the most common AMGs founded in viromes [35] favoring the nucleotide synthesis during viral replication [35]. In our study, we have also found an AMG related with lipopolysaccharide synthesis (Appendix A). In addition to AMGs in viruses, it has also been reported that some viruses also contain several microbial genes that compensate the infection forming branched pathways [36]. These microbial gene features truly present in viral genomes hinder bioinformatic tools to carry out a proper discrimination between viral genomes and microbial contamination [30,32].

Overall, none of the vSAGs sequenced in this study were present in the comprehensive IMG/VR v3.0 database, which contains more than 2 million viral contigs (Figure 2) [9] and 200,000 marine viral populations from different locations, including those used here in this SVG study [6,11]. Unfortunately, a CRISPR spacer–protospacer search using a comprehensive database containing more than 10 million spacers [37] did not shed any light into the virus–host interactions for the recovered deep-ocean viruses.

Metagenomic fragment recruitment is a common method to analyze, in silico, the relative abundance of viruses using viromes or cell metagenomes along different ocean sampling points [18,22]. Results (Figure 4) show dominant populations (vSAGs 88-3-B13 and 88-3-G20) of viruses obtained from the North Atlantic Ocean (MSP131 sampling site) that were widespread and abundant in the rest of ocean samples as well, despite being collected thousands of kilometers away and years apart (Figure 4). This is in accordance with previous works, in which a widespread common core set of viruses between remote bathypelagic regions was observed [22,34], which could be related to the global overturning circulation [26]. Despite the abundance of some of the deep-ocean viruses retrieved in this study by single-virus genomics, high-throughput viromics has not been able to “capture” these viral populations in previous surveys [6,11]. This result highlights the importance of SVGs to assemble putative ecologically relevant viral populations; for instance, as in the case of the microdiverse vSAG 37-F6-like pelagiphages, which are very abundant on the surface [18]. Our data also reveal the feasibility and potential of this technology. For a more complete discussion about limitations of this technology, we suggest a review on SVGs that addresses all pros and cons of this approach in comparison with viromics and other existing technologies [24]. Regarding some interesting advantages, when SVGs is combined with BONCAT-click technology, it can target and unveil the identity of active viral populations in a sample, since only active viruses are fluorescently labelled and, therefore, sorted and sequenced using the SVG pipeline [24]. In the same way, as for viromics and its advances over the last 20 years [37] SVGs has a promising perspective to complement existing viromic technologies that will aid in better deciphering the vast ocean virosphere.

## Figures and Tables

**Figure 1 viruses-14-01589-f001:**
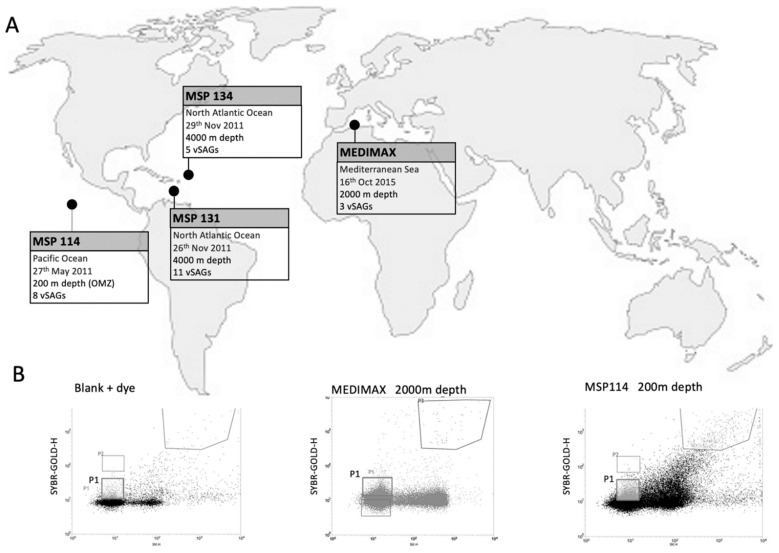
(**A**) Sampling information map. Location, date, depth and number of viral single amplified genomes (vSAGs) are indicated for each sampling point. (**B**) Flow cytometry sorting plots and gates for blanks (left panel; SYBR Gold-stained buffer TE), 2000 m depth MEDIMAX (middle panel), and MSP114 200 m depth (right panel; OMZ) samples. P1 gate, which shows the viral population, was employed to sort the single viruses employed in this study.

**Figure 2 viruses-14-01589-f002:**
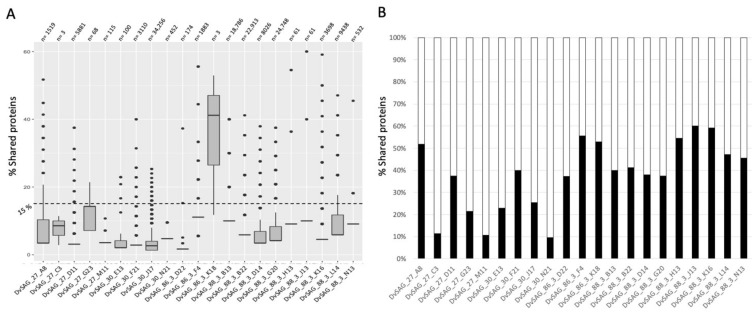
Novel deep-ocean viruses obtained from single-virus genomics. (**A**) Percentage of shared proteins (amino acid similarity >80% and query coverage >90%) between vSAGs and viruses from IMG/VR v3.0 database. “n” represents the number of known viruses sharing at least one protein with the corresponding vSAG. (**B**) Percentage of shared proteins for each vSAG versus its closest viral relative found in the IMG/VR v3.0 database. Detailed data about each comparison (e.g., number of proteins, similarity, coverage, and Scaffold ID for each similar virus found in the IMG/VR v3.0) can be found in the Appendix A).

**Figure 3 viruses-14-01589-f003:**
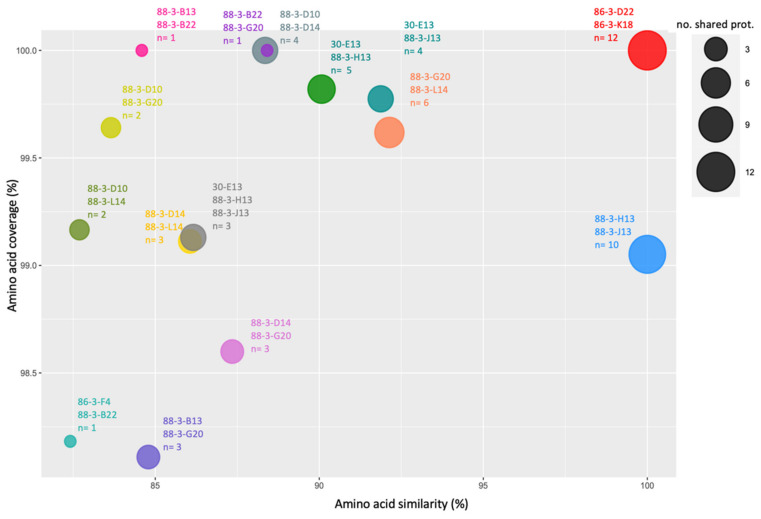
Shared proteins between vSAGs. Each circle represents two (or three) vSAGs sharing at least one protein. The area of the circle is related with the number of shared proteins.

**Figure 4 viruses-14-01589-f004:**
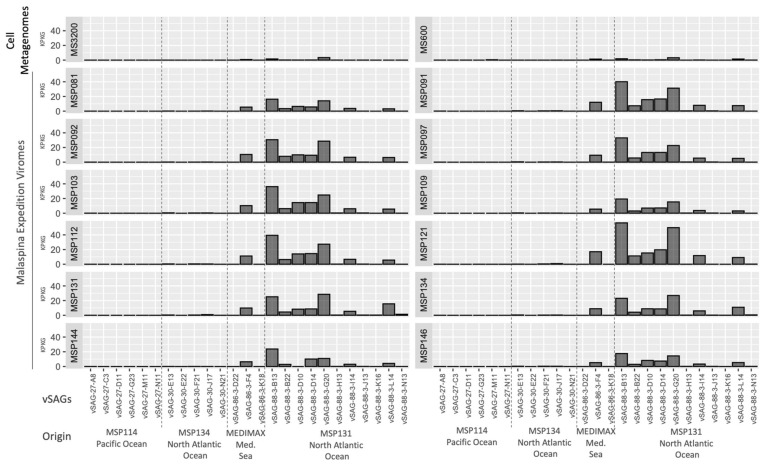
Abundance of vSAGs along different oceans. Abundance (as recruited kilobases per kilobases of vSAG genome length and gigabase of metagenome/virome, KPKGs) is calculated using two metagenomes (MS: Mediterranean Sea), and 12 bathypelagic viromes from the Malaspina expedition (MSP).

**Table 1 viruses-14-01589-t001:** Sample information.

Station	Zone	Depth (m)	vSAG Prefix	Coordinates	Sampling Date	Sorting Date	Preservation	WGA	% WGA	Sequenced	Characterized Viral Genomes
MSP 114	North Pacific Ocean	200 (OMZ)	27	14°31′43″ N 118°46′26″ W	27 May 2011	17 April 2015	GlyTE	92	27.71	17	8
MSP 131	North Atlantic Ocean	4000	88-3	17°25′39″ N 59°49′43″ W	26 November 2011	22 October 2015	GlyTE	17	5.12	12	11
MSP 134	North Atlantic Ocean	4000	30	18°19′38″ N 52°38′20″ W	29 November 2011	17 April 2015	GlyTE	46	13.86	16	5
MEDIMAX	Mediterranean Sea	2000	86-3	37°21′13″ N 00°17′10″ W	16 October 2015	22 October 2015	Fresh	162	48.80	15	3

vSAG prefix: Name prefix that indicates the original sample point of vSAG. Preservation: Preservation method of the sample from the sampling to the sorting process. WGA: Number of amplified single viruses. %WGA: Percentage of amplified single viral genomes. Sequenced: Number of sequenced single viral genomes. Characterized viral genomes: Bona fide identified viral genomes.

## Data Availability

Assembled genomes from analyzed vSAGs in this study are available in the JGI IMG/MER (Integrated microbial genomes and microbiomes, https://img.jgi.doe.gov/) under the IMG Genome IDs: 2818991326—2818991413.

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
