# Peer review of "Into the Dark: Exploring the Deep Ocean with Single-Virus Genomics"

_viruses, 2022, doi:10.3390/v14071589_

Round 1

Reviewer 1 Report

The presented manuscript describes a pilot study for single virus genomics from small volumes preserved marine samples. The manuscript provides an appropriate study design, well discussed findings and is well written. I was surprised there were ‘only’ 27 (or less, see comment below) viruses identified from ~1.3k sorted viruses. Although the authors touch this findings a little in the discussion, I would encourage them to discuss possible limitations in more detail.

Some specific comments:

Table 1: Please provide a description of e.g. the columns. the Table should be understood on its own without having to read the text

161-162: what has been done to assign taxonomy? Did you try vcontact2 or VICTOR?

Figure 2B: What does it exactly tell us? Do all vSAGs share protein similarity with the same virus mentioned in the text? Are these viruses from a database. If not, please add information about other ‘relatives’.

Figure S2 needs more explanation. The readers should be told what they see. Green = identity? ORFS for the different genomes? What happened with the gaps? Are they due to sequencing or due to alignment? Are the genomes 100% identical (no SNP)? If so, could it be that identical viruses have been sorted and/or sequenced in parallel? In that case, why would that figure/information be important?

Figure 3: typo in description. ‘protein’. Replace ‘Repts’ by ‘no. shared prot.’

Figure 4: typo in description?: ‘abundance is estimated’. Abundance of vSAGs related viruses since you performed analysis 80% similarity level. How is abundance calculated (y-axis)?

228: related to concerns above (fig S2). Are there 27 novel viruses if some of them are identical?

Author Response

REFEREE 1

We would like to thank reviewers and the editor for the time employed reading the text and for the interesting suggestions that they have provided. We have revised our manuscript according to those recommendations and included all changes and suggestions in this new version.

We are glad that referees have a positive opinion and find that “The manuscript provides an appropriate study design, well discussed findings and is well written” (Referee 1) and “The article presents novel and interesting data and is written well” (Referee 2)

The presented manuscript describes a pilot study for single virus genomics from small volumes preserved marine samples. The manuscript provides an appropriate study design, well discussed findings and is well written. I was surprised there were ‘only’ 27 (or less, see comment below) viruses identified from ~1.3k sorted viruses. Although the authors touch this findings a little in the discussion, I would encourage them to discuss possible limitations in more detail.

ANSWER: We would like to thank the referee for this comment. We have modified different parts of the manuscript to better discuss this point, providing to the readers, a more appropriate perspective according to the aim of the manuscript. However, limitation of this technology has been discussed in detail in a previous Nature Review Microbiology published by our group (“Single-Virus Genomics and Beyond”; Martinez-Martinez et al) which is  cited several times in the manuscript including now specifically in the discussion section. In this present study, because of budget constrain, and since it was originally conceived as a proof of concept of SVGs methodology applied to deep sea, we have sequenced a representative portion of the total sorted viruses (near to 20%). Obviously, this SVG data is complementary to viromic approach and does not aim to replace viromics. Indeed, this is very well discussed in our previous Nature Review Microbiology manuscript.  

Some specific comments:

Table 1: Please provide a description of e.g. the columns. the Table should be understood on its own without having to read the text

Done

161-162: what has been done to assign taxonomy? Did you try vcontact2 or VICTOR?

We have used the recently published viral sequence identification SOP with VirSorter2, CheckV and DRAMv: https://www.protocols.io/view/viral-sequence-identification-sop-with-virsorter2-5qpvoyqebg4o/v3?step=1.

This is a methodology employed by Sullivan´s Lab, which is one of the most relevant laboratories on marine virology. The employed version of VirSorter2 generates information related with viral genomic features, including structural, functional and taxonomy.

Figure 2B: What does it exactly tell us? Do all vSAGs share protein similarity with the same virus mentioned in the text? Are these viruses from a database. If not, please add information about other ‘relatives’.

We appreciate this comment. Legend of the figure has been modified and a new table (Supplementary table) has been added along with the information of all  vSAGs-close viral relatives found in the IMG/VR database.

Figure S2 needs more explanation. The readers should be told what they see. Green = identity? ORFS for the different genomes? What happened with the gaps? Are they due to sequencing or due to alignment? Are the genomes 100% identical (no SNP)? If so, could it be that identical viruses have been sorted and/or sequenced in parallel? In that case, why would that figure/information be important?

Legend of the figure and text of the main manuscript have been modified to improve the reader comprehension.

Figure 3: typo in description. ‘protein’. Replace ‘Repts’ by ‘no. shared prot.’

The figure and  description have been corrected.

Figure 4: typo in description?: ‘abundance is estimated’. Abundance of vSAGs related viruses since you performed analysis 80% similarity level. How is abundance calculated (y-axis)?

As explained in the new legend and also detailed in methods, abundance was estimated as the number of kilobases recruited per length of the vSAG (also in Kb) and divided per the virome/metagenome length in Gigabases (KPKGs). This has been extensively used in several leading publications in marine virology (Mizuno et al; Roux et al, Brum et al; Martine-Hernandez et al)

228: related to concerns above (fig S2). Are there 27 novel viruses if some of them are identical?

This sentence has been clarified.

Reviewer 2 Report

Review of the manuscript by Martinez-Hernandez et al., entitled “Into the dark: Exploring the deep ocean with single-virus genomics”, submitted to Viruses for consideration

General comment: The article presents novel and interesting data and is written well, even if some aspects must be improved for several aspects. I detail here below some comments that I hope can help the Authors in the revision of this work before publication.

Specific comments:

- The logic of the sampling and selection of the samples (or viruses) lost or not considered after each methodological step is not clear and should be fixed, also in the methods description. Only some of the sorted viruses (ca.1300) were MDA-amplified, and then some were randomly selected to be sequenced. In particular, why selecting randomly? Was it related to analysis failure? Budget limitation? Or, some other more scientific rationale related to the relevance/interest of samples?

- Not clear from the abstract that 60 sequenced single viruses resulted in 27 bona fide viruses (some resulted the same virus? Some were not successfully amplified? On the contrary, did some “single viruses” actually reveal contigs from more than a single virus? Please better explain).

- One part of the story tells that the 27 bona fide viruses have no similarity with viral sequences in the public databases, while the second part of the story shows good representation of these viruses in available viromes. I understand the work done by the Authors, but this apparently counterintuitive result should be better discussed to avoid misunderstanding.

- As the sequenced viruses appear novel, it would be expected to read of some effort by the Authors to exploit information from relevant oceanic MAGs or CRISPR-based analysis to try to infer putative hosts, if feasible. No information is provided on this aspect, which should be considered and fixed.

- I might have missed this, but were the metagenomes and viromes used for Fig.4, also assembled to get MAGs or viral genomes? This would be useful to get more insights on the 27 viruses found in this work.

- L43-63 this part of the introduction sounds not so fair… several papars using viromics have massively increased our current knowledge of the global viral diversity, which SVGs did not (or, much less)… So, I agree that these are complementary approaches as stated by the Authors, but this paragraph should be rephrased to better acknowledge the limits and possibilities with each of the two approaches, and why they are complementary. As Figure 4 also uses some metagenomes, something should be put in the introduction on the further utility/complementarity of metagenomics (also related to MAGs and CRISPRs, as commented above).

- L64 “Here, we use SVG to unveil the genome of 27 abundant deep ocean viruses from the”… this sentence is logically incorrect, as, a priori, the Authors could not know that they would have focused on 27 viruses. Please replhrase also based on my first specific comment

- L30 in addition to ref 3 on marine sediments, an appropriate general ref for viruses in the water column would be needed, for example seminal papers by Suttle, Fuhrman... for example Suttle, 2007 NatRevMicro Marine viruses-major players in the global ecosystem. L42, conversely, here an additional reference besides waters (Ref 12) would be needed for sediments, for example a relevant one showing higher quantitative role of viruses with increasing water depth may be Danovaro et al., 2015 AME Towards a better quantitative assessment of therelevance of deep-sea viruses, …

-Some simple word errors/repetitions should be fixed in several sentences, please pay particular attention before submitting

Author Response

REFEREE 2

We would like to thank reviewers and the editor for the time employed reading the text and for the constructive suggestions that they have provided. We have revised our manuscript according to those recommendations and incorporated all changes and suggestions of reviewers in this new version.

We are glad that referees have a positive opinion and find that “The manuscript provides an appropriate study design, well discussed findings and is well written” (Referee 1) and “The article presents novel and interesting data and is written well” (Referee 2)

Review of the manuscript by Martinez-Hernandez et al., entitled “Into the dark: Exploring the deep ocean with single-virus genomics”, submitted to Viruses for consideration

General comment: The article presents novel and interesting data and is written well, even if some aspects must be improved for several aspects. I detail here below some comments that I hope can help the Authors in the revision of this work before publication.

Specific comments:

- The logic of the sampling and selection of the samples (or viruses) lost or not considered after each methodological step is not clear and should be fixed, also in the methods description. Only some of the sorted viruses (ca.1300) were MDA-amplified, and then some were randomly selected to be sequenced. In particular, why selecting randomly? Was it related to analysis failure? Budget limitation? Or, some other more scientific rationale related to the relevance/interest of samples?

We really appreciate this valuable comment that allow us to improve the quality of our manuscript. Several detailed explanations have been added in the new version of the manuscript to help the readers better understand the results. In the same way we also address here these questions point by point:

Regarding the first question related with the number of amplified genomes:

  1. After sorting of viruses, next step in Single-virus genomics is the amplification of the genome of each single virus. As described in Martinez-Hernandez, F.; et al 2017, or Martínez-Martínez, J.; et al 2020, we employed a highly efficient and fidelity polymerase (Phi29 DNA polymerase) to perform a whole genome amplification (WGA) by multiple-displacement amplification (MDA reaction).

  1. Due to the delicate process to amplify the tiny amount of DNA enclosed in a single viral particle, MDA reaction doesn´t have an efficiency of 100%, and only the genome of some of the sorted events (individualized in a multiwell plate) are correctly amplified. But this is alsdo happening and affecting to single cell genomi fields. Not all sorted cells can be amplified because of several biases (inefficient cell lysis, etc…). In this paper the MDA successful rate was as a mean 24%, which in general is a good efficiency in comparison even with single-cell genomics.

  1. This bias is associated to any kind of single-cell genomics process (prokaryotic or eukaryotic cells), not only in viruses. There are many reviews on that by Quake and collaborators, Stepanauskas´s laboratory, Woyke´s lab at JGI, etc…

Regarding the number of selected amplified genomes to sequence:

  • The main aim of this work was to demonstrate the feasibility of the Single-virus genomics to recover the genome of viruses from deep ocean samples that normally require of large volume samples difficult to collect at 4,000 m. , or that in some cases are long-time  preserved, such as our case. As discussed in the last part of the manuscript, we also address the issue of best way to preserve these kind of samples, which is in general important in virology.

  • In this proof of concept with deep ocean samples (most of them collected several years before to the first successfully demonstration of SVGs methodology applied to environmental samples), we do not intend to carry out a massive sequencing because of several reason (e.g. budget constrain). For this reason, we sequenced a representative sub-sampling (about 20%) of the positive amplified genomes.

  • As clarified in the paper, after this valuable comment of the referee, SVGs is a complementary method to the most conventional method today, which obviously is viromics. This methodology doesn´t pretend to compete with the bulk sequencing of genomes that current viromics can generate. Instead, as it has been demonstrated in this or in other previous works (as Martinez-Hernandez, F.; et al 2017 or de la Cruz Peña, MJ,; et al 2018), SVGs complement viromic data because can recover the genome of ecologically relevant viral populations that in some cases are overlooked by viromics (even after recovering hundreds of thousands of genomes from the same sample; see for instance our Nature Microb Review by Martinez-Martinez et al or discussion in Martinez-Hernandez et al 2017 Nat Comm).

- Not clear from the abstract that 60 sequenced single viruses resulted in 27 bona fide viruses (some resulted the same virus? Some were not successfully amplified? On the contrary, did some “single viruses” actually reveal contigs from more than a single virus? Please better explain).

Thanks for the comment. We have clarified this result in the abstract and in the main test.

After sequencing 60 of the single amplified genomes, we performed an strict pipeline using VirSorter2, checkV and DRAMv (detailed in methods) to discriminate if these genomes matched as viruses or not. Specific viral protein annotation allowed us to clasify 27 of them as bona-fide viral genomes, thanks to the presence of specific viral proteins as terminases, phage lysozymes or structural viral proteins.

We also discussed this point in the third paragraph of the discussion section.

- One part of the story tells that the 27 bona fide viruses have no similarity with viral sequences in the public databases, while the second part of the story shows good representation of these viruses in available viromes. I understand the work done by the Authors, but this apparently counterintuitive result should be better discussed to avoid misunderstanding.

This is an important comment. We have clarified this idea in the manuscript with a new paragraph in the discussion addressing this issue. Indeed, this point supports the previous results observed in our first application of SVGs in surface seawater (Martinez-Hernandez, F.; et al, 2017), in which we recovered, by the first time, a highly abundant and widespread virus not present in different assembled viromes despite the different metagenomic efforts.

- As the sequenced viruses appear novel, it would be expected to read of some effort by the Authors to exploit information from relevant oceanic MAGs or CRISPR-based analysis to try to infer putative hosts, if feasible. No information is provided on this aspect, which should be considered and fixed.

We appreciate this comment. Indeed, although was not originally mentioned in the text, we performed a virus-host search by searching for matchs in CRISPR-spacer-protospacer. For that, we used a huge database recently released containing more than 11 millions of spacers for prokaryotes (see paper by Moïra et al 2021 “Streamlining CRISPR spacer-based bacterial host predictions to decipher the viral dark matter”), but unfortunately we have not found any match. Thus, there is no room for more speculation. In addition, MAG search based on homology have not clarified anything. In both cases, it can be easily explained because  reference prokaryote genomes from the deep ocean is clearly under-represented in databases, and thus it is very complicated to find a match.   

- I might have missed this, but were the metagenomes and viromes used for Fig.4, also assembled to get MAGs or viral genomes? This would be useful to get more insights on the 27 viruses found in this work.

Viromes from Malaspina expeditions were analyzed and assembled by the first time as part of the Global Ocean Virome (GOV v.2) in the work Roux, S.; et al 2016. (Ecogenomics and potential biogeochemical impacts of globally abundant ocean viruses). This data was re-analyzed in a more comprehensive work (Gregory A.C,; et al 2019, Marine DNA viral Macro- and Microdiversity from pole to pole) in which near 200,000 viral contigs (n=195,728) were assembled. All these viral contigs (obtained from Malaspina expedition and other worldwide expeditions) are included in the IMG/VR viral database v.3 employed in this work. Nevertheless, as our results show, viral genomes recovered by SVGs were not assembled in that metagenomic data, although, as the fragment recruitment indicate, they are present in these viromes. We have included a paragraph at the end of the discussion highlighting this situation.

- L43-63 this part of the introduction sounds not so fair… several papars using viromics have massively increased our current knowledge of the global viral diversity, which SVGs did not (or, much less)… So, I agree that these are complementary approaches as stated by the Authors, but this paragraph should be rephrased to better acknowledge the limits and possibilities with each of the two approaches, and why they are complementary. As Figure 4 also uses some metagenomes, something should be put in the introduction on the further utility/complementarity of metagenomics (also related to MAGs and CRISPRs, as commented above).

This part of the introduction has been modified to better represent the importance of the viromic. As  described in the new paragraph, we are aware of the relevance of viromic methodologies, not only to recover viral genomes. Thanks to the comment of the referee, introduction better represents our opinion. We think SVG complements viromics but never can replace viromics.  

- L64 “Here, we use SVG to unveil the genome of 27 abundant deep ocean viruses from the”… this sentence is logically incorrect, as, a priori, the Authors could not know that they would have focused on 27 viruses. Please replhrase also based on my first specific comment

We have modified this sentence.

- L30 in addition to ref 3 on marine sediments, an appropriate general ref for viruses in the water column would be needed, for example seminal papers by Suttle, Fuhrman... for example Suttle, 2007 NatRevMicro Marine viruses-major players in the global ecosystem. L42, conversely, here an additional reference besides waters (Ref 12) would be needed for sediments, for example a relevant one showing higher quantitative role of viruses with increasing water depth may be Danovaro et al., 2015 AME Towards a better quantitative assessment of therelevance of deep-sea viruses, …

Both references have been added.

-Some simple word errors/repetitions should be fixed in several sentences, please pay particular attention before submitting

Typo errors have been fixed

Reviewer 3 Report

Although the authors argue for the merits of SVG, and even though it is a novel technique which does offer another tool with which to investigate viral diversity, here they only managed to add 27 partial genomes to the literature. I'd like to see a more balanced criticism of "classic" shotgun viral metagenomes versus SVG throughout the manuscript, as the former is arguably solely responsible for the 2+ million virus genomes in the IMG/VR database, compared to SVG which can be very hit and miss and cost a lot of money for very little return. 

The authors make the statement that SVG is able to "recover novel abundant and representative viruses not previously sequenced by other methodologies" based on 1ml of seawater and a total of 27 partial viral  genomes. I would be far more circumspect in the conclusions drawn from such a dataset.

SVG is a very sensitive technique as one is trying to isolate and amplify an extremely small nucleic acid sample. Did the authors sequence controls to account for contamination from processing or the lab environment? 

L18: I would change "Protein annotation" to "Genome annotation"

L33: "have broaden" change to "have broadened"

L34: "Nevertheless, the huge extant" change to "Nevertheless, the full extant"

L37: "The marine environment has been one of" change to "The marine environment is one of"

L37-L39 - I would split this sentence into two sentences.

There are several quirks about the use of English in the manuscript that need to be addressed. Please seek out someone who's first language is English to read through the manuscript and help correct these.

L99: "concentred"

Why is the sorting of viruses into pools of 10 or 20 mentioned in the methods when these results aren't reported?

L113: I would change the title of the section to "Sequencing, viral identification and fragment recruitment analysis"

L115: "(2 x250)" change to "(2 x 250bp)"

L126: "blastp" change to "BLASTp"

L138: "also employed to relative estimate the viral abun-" change to "also employed to estimate the relative viral abun-". There are many of these examples throughout the text, which is why I encourage the authors to employ a language/text editing service.

L154: "checkV" change to "CheckV"

L154: "allowed to confidently" change to "allowed us to confidently"

L159-162: Could another reason for the lack of classification be the partial nature of the genomes recovered? In other words you may be missing key proteins that could lead to classification? I would include this as a possible reason for not being able to classify the viruses.

L166-167: Should it not be <80% and <90% or are you reiterating the cutoff values? If it is the latter, I would remove the statement in brackets as this is already mentioned in the methods.

L180: "pair-by-pair" change to "pairwise". "vSAGss" change to "vSAGs"

L181: You again reiterate the cutoff values, which is unnecessary.

L192: "preotein" change to "protein"

L197: "An homogeneous" change to "A homogeneous"

L205: "Abundance is stimated" change to "Abundance estimated"

L212: I think the statement that the authors were able to sort "hundreds of viruses" is disingenuous. They managed to sort 1328 virus like particles (VLP), and only 60 were sequenced (Well below a hundred. This also means that the identity of the remainder of the particles could not even be established) and only 27 of which they confirmed to be bona fide virus genomes.

L217: "method" change to "methods"

L218: Start the sentence with "The"

L220: Add a percentage sign after "15.56"

Author Response

REFEREE 3

We would like to thank reviewers and the editor for the time employed reading the text and for the constructive suggestions that they have provided. We have revised our manuscript according to those recommendations and incorporated all changes and suggestions of reviewers in this new version.

We are glad that referees have a positive opinion and find that “The manuscript provides an appropriate study design, well discussed findings and is well written” (Referee 1) and “The article presents novel and interesting data and is written well” (Referee 2)

Although the authors argue for the merits of SVG, and even though it is a novel technique which does offer another tool with which to investigate viral diversity, here they only managed to add 27 partial genomes to the literature. I'd like to see a more balanced criticism of "classic" shotgun viral metagenomes versus SVG throughout the manuscript, as the former is arguably solely responsible for the 2+ million virus genomes in the IMG/VR database, compared to SVG which can be very hit and miss and cost a lot of money for very little return.  The authors make the statement that SVG is able to "recover novel abundant and representative viruses not previously sequenced by other methodologies" based on 1ml of seawater and a total of 27 partial viral  genomes. I would be far more circumspect in the conclusions drawn from such a dataset.

We would like to thank this comment to the referee, since it is useful to better balance some parts of the manuscript. We have rewritten some paragraphs to better provide an idea of the usefulness of SVG (see for instance discussion) in comparison with viromics. We never aim that SVG replaces viromics, and we never say that SVG is better than any other technique. We believe that SVG, as any other technique, complement data obtained from other approaches and it has its own pros and cons. Several techniques are complementary and can be joined together to better answer biological questions. It seems that the referee got the sense that we are kind of “arguing against viromics”. Never!. We totally agree that the low number of recovered viral genomes in this work are not comparable with the ability of viromic methodologies to recover huge number of viral genomes. But, as we have clarified in the manuscript (thanks to the observation by this referee) the main target of this work is to demonstrate the feasibility of the Single-virus genomics to work with samples recovered from these environments. This proof of concept is a starting point to future applications and improvements of single-virus (and related methodologies). As discussed in previous articles, even using high-throughput conventional viromics, we have missed important “viral pieces” in nature. See for more discussion  our recent manuscript published in Nature Reviews Microbiology, Martínez-Martínez, et al., 2020). In the same way as viromics has evolved over the last 20 years, maybe SVG has also some avenues to explore, like the combination of BONCAT-click and SVG to target only active viruses recently released in a community.  

SVG is a very sensitive technique as one is trying to isolate and amplify an extremely small nucleic acid sample. Did the authors sequence controls to account for contamination from processing or the lab environment? 

As discussed in several leading papers on  single cell genomics by different labs (e.g JGI, Stepanauskas´s laboratory, etc..), we also have a database containing sequences obtained from negative amplified controls, that we use to identify putative contaminations related with all the process of SVGs. This has been previously discussed in (Martinez-Hernandez et al 2017, Martinez-Martinez et al 2020). Sequences characterized as bona-fide vSAGs in this work didn´t match obviously  with these negative controls, which by the way, are typical contaminants from molecular lab reagents that do not have anything to do with viral gene features.  

L18: I would change "Protein annotation" to "Genome annotation"

Done

L33: "have broaden" change to "have broadened"

Done

L34: "Nevertheless, the huge extant" change to "Nevertheless, the full extant"

Done

L37: "The marine environment has been one of" change to "The marine environment is one of"

Done

L37-L39 - I would split this sentence into two sentences.

Done

There are several quirks about the use of English in the manuscript that need to be addressed. Please seek out someone who's first language is English to read through the manuscript and help correct these.

L99: "concentred"

Done

Why is the sorting of viruses into pools of 10 or 20 mentioned in the methods when these results aren't reported?

We use these pools as positive controls during MDA. This is very common in single cell and virus genomics approaches, since these well represent internal positive controls to check that sorting was performed in a good manner. These wells, as they contain dozens of single-cells (1-fold more than a single virus), amplification during MDA should be always positive.

L113: I would change the title of the section to "Sequencing, viral identification and fragment recruitment analysis"

Done

L115: "(2 x250)" change to "(2 x 250bp)"

Done

L126: "blastp" change to "BLASTp"

Done

Round 2

Reviewer 3 Report

Figure 1 legend: Which gates were used to collect the vSAGs, P1, P2 or P3? The image is not very clear. Could the authors provide a better resolution image?

Line 300: Change “which makes very difficult a proper identification and characterization of novel viruses” to “which makes the proper identification and characterization of novel viruses very difficult”

Line 303: Change favorizing to favoring

Line 325: Change “despite they were collected thousands of kilometers away, and years apart” to “despite being collected thousands of kilometers away, and years apart”

Line 329-335: The authors state that: “high throughput viromics has not been able to “capture” these viral populations in previous surveys” and then go on to use the example of “vSAG 37-F6-like pelagiphages, which are very abundant in surface”. Do the authors really expect that classic viral metagenomics would not be able to detect this viral species in bathypelagic samples, but that SVG would? Surely it’s just a matter of sequencing depth?

Line 334: Change “assembly” to “assemble”

Author Response

Dear editors,

We appreciate the time of editors and reviewers to improve our manuscript. We believe, this updated version of the manuscript is now suitable for publication and include all suggestions raised by all reviewers in first and second round of review.

Sincerely

Manuel Martinez Garcia

Reply to referees

Line 35, There should be reference numbers 6 to 8 in "the auxiliary metabolic
genes (AMGs) . Over the last years".

  • Done

Line 137, "(2 x250 bp)" should be "(2 x 250 bp)" (a space in front of "250"
is added) as already mentioned by Reviewer #3.

  • Done

"BLASTp" is not commonly used. It should be "BLASTP" as used in the BLAST
manual and paper.

  • Done

Line 166, is "North Pacific (4,000 m depth) Oceans" correct? Figure 1 and
Table 1 show that the depth of this site was 200 m.

  • Thanks for this correction. It is 200 m (OMZ) as detailed in the rest of the manuscript.

All abbreviations should be explained the first time it is used in the text.
For example, viral single amplified genomes (vSAGs). In addition,
abbreviations should be consistent over the text, e.g., "Fig. 2" and "Fig 2"
should be same.

  • Thanks for the comment. We have reviewed all the abbreviations.

In the abstract appears “… the uncultured virus vSAG 37-F6”. Here vSAG is part of the name, but this abbreviation is explained in the main text.

Line 280, is "interpedently" correct?

  • Thanks, it should be: “independently” (Modified)

Figure 1 legend: Which gates were used to collect the vSAGs, P1, P2 or P3? The image is not very clear. Could the authors provide a better resolution image?

- Thanks for the comment. We have clarified which gate (P1) was employed to sort these viruses in this study. A better resolution figure has been submited. 

Line 300: Change “which makes very difficult a proper identification and characterization of novel viruses” to “which makes the proper identification and characterization of novel viruses very difficult”

  • Done

Line 303: Change favorizing to favoring

  • Done

Line 325: Change “despite they were collected thousands of kilometers away, and years apart” to “despite being collected thousands of kilometers away, and years apart”

  • Done

Line 334: Change “assembly” to “assemble”

  • Done

Line 329-335: The authors state that: “high throughput viromics has not been able to “capture” these viral populations in previous surveys” and then go on to use the example of “vSAG 37-F6-like pelagiphages, which are very abundant in surface”. Do the authors really expect that classic viral metagenomics would not be able to detect this viral species in bathypelagic samples, but that SVG would? Surely it’s just a matter of sequencing depth?

Thanks for this comment, which we really think it can help us to improve the message of our work. It is not only sequencing depth but also how much microdiversity is in it. In fact, this has been demonstrated in previous publications (Martinez-Martinez et al, 2020 Nat Rev Microb; Martinez-Garcia et al 2017, Nat Comm and other authors as well including by Simon Roux, deCarcer et al , etc…) and discussed in a recent Nature Review Microbiology (“Single Virus Genomics and beyond”). We have discussed this issue in the paper citing a few examples using single virus genomics in parallel with viromics. For instance see lanes: Lane 67: For instance, a large sample volume (order of liters) is required, and in some cases, the recovery of microdiverse viral populations (diversity within the same viral species) could be hindered by inherent problems during metagenomic assembly [10,18,21]. Lane 531: The application of SVGs in surface seawater in a prior study highlighted the limitation of viromics to assembly microdiverse viral populations [18,21],

Indeed, some/many vSAGs originally discovered in surface by single virus genomics (Martinez-Hernandez et al 2017, Nat Comm) are highly abundant and widespread in the current viromes and completely covered by their reads, but however, metagenomic assembly failed to recover the genomes of these microdiverse viral populations. In that same paper, the last main figure addressed that specific issue and we explained why they could not be assembled. Basically,  assemblers failed because of the high microdiversity within these viral populations. Indeed, we performed simulations to corroborate our findings. Both, empirical data and in silico simulations indicated that microdiversity hinders metagenomic assembly. There are also other recent examples in Microbiology from other ecosystems, such as Ramos-Barbero et al (“The good, the bad and the ugly”) in crystallizer ponds.